# Asymptomatic carriage of intestinal protists is common in children in Lusaka Province, Zambia

Mable Mutengo[1,2‡]*, Michaela Kaduková[3,4‡], Namwiinga R. Mulunda[1,2], Freeman W. Chabala[1], Alejandro Dashti[4], Kyoko Hayashida[5], Stanley Chinyanta[1], Kelly Chisanga[1], Lourdes Castro[4], Sergio Sánchez[4], James Mwansa[6], Pamela C. Köster[4,7,8], David González-Barrio[4], Jenny G. Maloney[9], Mónica Santín[9], Javier Sotillo[4], David Carmena[4,10]*

**1** Institute of Basic and Biomedical Sciences, Levy Mwanawasa Medical University, Lusaka, Zambia, **2** Department of Pathology and Microbiology, University Teaching Hospitals, Lusaka, Zambia, **3** Department of Epizootiology, Parasitology and Protection of One Health, University of Veterinary Medicine and Pharmacy in Košice, Košice Slovakia, **4** Parasitology Reference and Research Laboratory, Spanish National Centre for Microbiology, Health Institute Carlos III, Majadahonda, Madrid, Spain, **5** International Institute for Zoonosis Control, Hokkaido University, Sapporo, Japan, **6** Directorate of Research and Postgraduate Studies, Lusaka Apex Medical University, Lusaka, Zambia, **7** Women for Africa Foundation, Madrid, Spain, **8** Faculty of Health Sciences, Alfonso X El Sabio University (UAX), Villanueva de la Cañada, Spain, **9** Environmental Microbial and Food Safety Laboratory, Agricultural Research Service, United States Department of Agriculture, Beltsville, Maryland, United States of America, **10** Centre for Biomedical Research Network (CIBER) in Infectious Diseases, Health Institute Carlos III, Madrid, Spain

‡ These first authors contributed equally to this article.
* mablem38@gmail.com (MM); dacarmena@isciii.es (DC)

**Data Availability Statement:** The data that supports the findings of this study are available within the main body of the manuscript and its supplementary material. Sequences obtained in

## Abstract

### Background

PCR-based screenings on the presence of diarrhoea-causing intestinal protist species are limited in Zambia, resulting in inaccurate current prevalence and epidemiological data. Sensitive PCR-based methods are particularly well suited for detecting subclinical infections in apparently healthy carriers.

### Methodology

In this prospective cross-sectional study, we investigated the occurrence of the most common intestinal protists in an apparently healthy paediatric population (5–18 years) in Lusaka Province, Zambia. We collected single stool samples ($n = 256$) and epidemiological questionnaires on demographics, behavioural habits, drinking water and toilet access from participating children. We used PCR for the initial screening of samples for the presence of intestinal protist species and Sanger and next-generation sequencing for genotyping. We conducted statistical analyses to assess the association of the gathered variables with an increased likelihood of the investigated pathogens.

this study were deposited in GenBank under accession numbers PQ185660–PQ185667 and PQ213645–PQ213681 (*Giardia duodenalis*), PQ191442–PQ191445 (*Cryptosporidium* spp.), and PQ200214–PQ200222 and PQ336781-PQ376832 (*Blastocystis* sp.).

**Funding:** This study was partially funded by the Health Institute Carlos III, Spanish Ministry of Economy and Competitiveness, (PI16CIII/00024 to DC) and by the USDA-ARS (8042-32000-112-00-D to MS). The Article Processing Charge for this publication was covered by Women for Africa Foundation and Eurofins. The funders had no role in study design, data collection and analysis, decision to publish, or preparation of the manuscript.

**Competing interests:** The authors have declared that no competing interests exist.

## Principal findings

*Blastocystis* sp. was the most prevalent intestinal protist found (37.9%, 97/256; 95% CI: 31.9–44.1), followed by *Giardia duodenalis* (30.9%, 79/256; 95% CI: 25.3–36.90), *Entamoeba dispar* (13.3%, 34/256; 95% CI: 9.4–18.1), and *Cryptosporidium* spp. (4.3%, 11/256, 95% CI: 2.2–7.6). *Entamoeba histolytica* was not detected. Based on Sanger sequencing results, subtypes ST2 (44.3%, 43/97), ST1 (35.1%, 34/97), and ST3 (20.6%, 20/97) were identified within *Blastocystis* sp. and assemblages B (71.0%), A+B (16.1%), and A (12.9%) within *G. duodenalis*. *Cryptosporidium parvum* (81.8%) and *C. hominis* (18.2%) were the only two *Cryptosporidium* species found. Living in the Kafue District was positively associated with higher infection rates by *G. duodenalis* and *Blastocystis* sp. Schoolchildren living in Chongwe District were more likely to be infected by *Cryptosporidium* spp.

## Conclusions/Significance

Intestinal protist infection/colonization is a common finding in apparently healthy children in Lusaka Province, Zambia. Asymptomatic carriers may play an underestimated role as spreaders of gastrointestinal parasitic infections. This study improves our current understanding of the epidemiology of diarrhoea-causing protists in Zambia and sub-Saharan Africa and indicates that the role of asymptomatic carriers of gastrointestinal parasites in transmission should be further explored.

### Author summary

Intestinal protist parasites are a major public health concern in sub-Saharan countries with limited access to safe drinking water and with poor sanitation infrastructures. Children are at higher risk of infection because of suboptimal personal hygiene practices and immature immune systems. Asymptomatic carriers of intestinal protists were common in children in Lusaka Province, Zambia and their role on the epidemiology and human-to-human spread of these pathogens may have been underestimated. In Zambia, all available epidemiological information on these pathogens comes from Lusaka Province, but virtually nothing is known from other geographical regions. More research should be conducted in these areas to ascertain the true epidemiology and public health impact on these pathogens in the country.

## 1. Introduction

Cryptosporidiosis (*Cryptosporidium* spp.), giardiasis (*Giardia duodenalis*), and amoebiasis (*Entamoeba histolytica*) are important intestinal protozoan infections, causing significant morbidity and mortality particularly in young children living in limited-resource settings. In sub-Saharan African and southeast Asian countries, the Global Enteric Multicenter Study (GEMS) has identified cryptosporidiosis as the second cause of childhood mortality after rotaviral infection, and amoebiasis as an increased risk of death in infants and toddlers with moderate-to-severe diarrhoea [1]. Similarly, the Malnutrition and Enteric Diseases (MAL-ED) multicenter study has linked giardiasis with malnutrition, growth retardation, and impaired cognitive development during early childhood [2,3]. On the other hand, the stramenopile *Blastocystis* sp.

is an intestinal microeukaryote of uncertain pathogenicity whose epidemiology in sub-Saharan Africa is still poorly understood. Although some clinical studies have associated the presence of *Blastocystis* with intestinal [4] and extra-intestinal [5] disorders, recent large metagenomics analyses have revealed that the protist may be an indicator of healthier diets and favourable cardiometabolic outcomes [6]. These findings have led some authors to suggest a change of paradigm in which *Blastocystis* could be seen as a potentially beneficial microorganism rather than a harmful intestinal parasite [7].

All the above-mentioned protist species are faecal-orally transmitted indirectly through the consumption of water or food contaminated with faecal matter or directly through contact with infected humans or animals. Therefore, living in settings with limited or no access to safe drinking water and poor sanitation infrastructures and being in close contact with livestock are factors positively associated with an increased likelihood of carrying intestinal protists [8,9].

PCR-based molecular tools coupled with Sanger or next-generation sequencing methods have significantly improved our understanding of the epidemiology of diarrhoea-causing intestinal protists as they allow for i) highly sensitive and specific pathogen detection, ii) differential diagnosis of species whose developmental stages are morphologically undistinguishable, iii) detection of rare/underrepresented species/genotypes, and iv) conducting epidemiological and outbreak investigations to ascertain sources of infection, transmission pathways, and zoonotic potential.

The *Cryptosporidium* genus encompasses at least 47 valid species and more than 120 genotypes of uncertain taxonomic position [10–13]. Over 21 *Cryptosporidium* species and 2 genotypes have been reported in humans [10–14], with *C. hominis* and *C. parvum* accounting for *ca.* 90% of the human cryptosporidiosis cases documented globally, including Africa [15]. *Giardia duodenalis* is regarded as a species complex comprising eight (A-H) assemblages with marked differences in host range and specificity [10]. Although most human cases of giardiasis are caused by zoonotic assemblages A and B, sporadic infections by host-adapted assemblages C/D (canids), F (felids), and E (ruminants) have also been documented primarily in children and immunocompromised individuals [10]. Regarding *E. histolytica*, this species displays a relatively low level of nucleotide diversity across its genome, so no routine genotyping schemes are currently in place for this pathogen [16]. However, because *E. histolytica* cysts and trophozoites are morphologically identical to those of non-pathogenic *Entamoeba* species (i.e., *E. dispar*, *E. moshkovskii*, and *E. bangladeshi*), a molecular approach is required for the correct differential diagnosis of pathogenic *E. histolytica*. Finally, *Blastocystis* is divided into 44 subtypes (STs) [17–19], and 14 have been reported in both animals and humans [20–22].

In Zambia, the presence of diarrhoea-causing intestinal protists has been investigated in patients with human immunodeficiency virus (HIV) and paediatric and adult populations with and without clinical manifestations. Most of these surveys were conducted by conventional (light/immunofluorescence microscopy) or immunoenzymatic (ELISA) methods (Table 1).

Reported prevalence rates ranged from 2–54% for *Blastocystis* sp., 2–31% for *Cryptosporidium* spp., 1–12% *Cystoisospora belli*, 2% for *E. histolytica*, and 1–28% for *G. duodenalis*. Few studies have been conducted on livestock (cattle, pigs) and environmental (water) samples. PCR-based studies are limited, so information on the frequency and molecular diversity of these pathogens (particularly for *G. duodenalis* and *Blastocystis* sp.) is scarce and highly needed. In addition, most of the surveys conducted were geographically restricted to the Lusaka province, meaning that the epidemiological scenario shown in Table 1 may differ in other Zambian regions.

To overcome some of the knowledge gaps stated above, we used molecular methods to assess the frequency and genetic diversity of *G. duodenalis*, *Cryptosporidium* spp., *Entamoeba*

**Table 1. Occurrence and genetic diversity of intestinal protists in humans and domestic animals, Zambia, 1989–2022.**

| Pathogen | Population | Clinical symptoms | Samples (n) | Region | Detection method | Infection rate (%) | Species/Genotypes/Subtypes (n) | Reference |
|---|---|---|---|---|---|---|---|---|
| *Blastocystis* sp. | HIV+ | Yes | 90 | Lusaka | CM | 21.1 | – | [23] |
| | Adult | No | 105 | Lusaka | CM | 21.0 | – | [23] |
| | HIV+ | Yes | 242 | Lusaka | CM | 1.7 | – | [24] |
| | Paediatric (HIV+) | Yes | 106 | Lusaka | CM | 1.9 | – | [25] |
| | Paediatric | Yes | 90 | Lusaka | CM | 2.2 | – | [25] |
| | Paediatric | Yes | 93 | Namwala | CM | 53.8 | – | [26] |
| | Clinical | NS | 85 | Lusaka | PCR | –[a] | ST1 (9), ST2 (7), ST3 (11), ST6 (1) | [27] |
| *Cryptosporidium* spp. | HIV+ | Yes | 90 | Lusaka | CM | 2.2 | – | [23] |
| | HIV+ | Yes | 242 | Lusaka | CM | 3.3 | – | [24] |
| | Paediatric (HIV+) | Yes | 44 | Lusaka | CM | 13.6 | – | [28] |
| | Paediatric | Yes | 134 | Lusaka | CM | 6.0 | – | [28] |
| | Adult | Yes | 162 | Lusaka | CM | 10.0 | – | [29] |
| | Paediatric | Yes | 222 | Lusaka | CM | 17.6 | – | [30] |
| | Paediatric (HIV+) | Yes | 106 | Lusaka | CM | 28.3 | – | [25] |
| | Paediatric | Yes | 90 | Lusaka | CM | 18.9 | – | [25] |
| | Paediatric | Yes | 93 | Namwala | CM, IFA, FISH | 8.6 | *C. parvum* (8) | [26] |
| | Farm workers | No | 289 | Lusaka | ELISA, PCR | 6.2 | *C. parvum* (12); *C. hominis* (3) | [31] |
| | Paediatric | No | 403 | Kafue | IFA | 28.0 | – | [32] |
| | Paediatric | No | 786 | Kafue | IFA | 30.7[K] | – | [33] |
| | Clinical | Yes | 71 | Lusaka | PCR | –[a] | *C. hominis* (42), Ia (12), Ib (8), Id (2), Ie (20); *C. parvum* (27), IIc (12), IIe (14), IIs (1); *C. felis* (1); *C. meleagridis* (1) | [34] |
| | HIV+ | NS | 326 | Namwala | IFA | 9.5 | – | [35] |
| | Paediatric | Yes | 490 | Various | CM, PCR | 10.2 | *C. hominis* (14), Ia (6), Ib (7), Ie (1); *C. parvum* (2), IIc (2) | [36] |
| | Dog | No | 390 | Lusaka | CM | 5.9 | – | [37] |
| | Cattle | No | 740 | Various | CM, PCR | 19.2 | *C. bovis* (14); *C. parvum* (28); *C. suis* (1) | [38] |
| | Cattle | No | 207 | Lusaka | ELISA, PCR | 33.8 | *C. parvum* (13); *C. bovis* (7); *C. ryanae* (1)[c] | [31] |
| | Cattle | – | 328 | Monze, Mumbwa, Lusaka | CM | 14.5 | ND | [43] |
| | Sheep | – | 190 | Monze, Mumbwa, Lusaka | CM | 5.3 | ND | [43] |
| | Goat | – | 245 | Monze, Mumbwa, Lusaka | CM | 1.2 | ND | [43] |
| | Pig | No | 217 | Lusaka | IFA | 44.2 | – | [39] |
| | Flies | – | 17 | Lusaka | qPCR | 5.9 | ND | [40] |
| | Water | – | 21 | Lusaka | CM, IFA | 95.2 | – | [29] |
| | Water | – | 40 | Lusaka | qPCR | 7.5 | ND | [40] |
| *Cystoisospora belli* | HIV+ | Yes | 90 | Lusaka | CM | 7.7 | – | [23] |
| | Paediatric | Yes | 178 | Lusaka | CM | 0.6 | – | [28] |
| | Adult | Yes | 162 | Lusaka | CM, IFA | 12.0 | – | [29] |

*(Continued)*

**Table 1.** (Continued)

| Pathogen | Population | Clinical symptoms | Samples (n) | Region | Detection method | Infection rate (%) | Species/Genotypes/Subtypes (n) | Reference |
|---|---|---|---|---|---|---|---|---|
| | Paediatric (HIV+) | Yes | 106 | Lusaka | CM | 1.9 | – | [25] |
| | Paediatric | Yes | 90 | Lusaka | CM | 2.2 | – | [25] |
| *Entamoeba histolytica* | HIV+ | Yes | 242 | Lusaka | CM | 2.1 | – | [24] |
| | Paediatric (HIV+) | Yes | 44 | Lusaka | CM | 2.3 | – | [28] |
| *Giardia duodenalis* | HIV+ | Yes | 242 | Lusaka | CM | 2.1 | – | [24] |
| | Paediatric (HIV+) | Yes | 44 | Lusaka | CM | 2.3 | – | [28] |
| | Paediatric (HIV+) | Yes | 106 | Lusaka | CM | 3.8 | – | [25] |
| | Paediatric | Yes | 90 | Lusaka | CM | 7.8 | – | [25] |
| | Paediatric | Yes | 93 | Namwala | CM, IFA, FISH | 19.4 | – | [26] |
| | Paediatric | No | 403 | Kafue | IFA | 28.0 | – | [32] |
| | Paediatric | No | 786 | Kafue | IFA | 28.0[b] | – | [33] |
| | Paediatric | No | 329 | Lusaka | CM, PCR, PCR-RFLP | 10.0 | AII (9), BIII (4), BIV (17), BIII+BIV (3) | [41] |
| | Cattle | No | 377 | Chilanga/ Lusaka | ELISA | 34.5 | – | [42] |
| | Pig | No | 217 | Lusaka | IFA | 12.0 | – | [39] |

CM: Conventional microscopy; ELISA: Enzyme-linked immunosorbent assay; FISH: Fluorescent in situ hybridization; HIV: Human immunodeficiency virus; IFA: Immunofluorescent antibodies; ND: Not determined; NS: Not specified; PCR: Polymerase chain reaction; RFLP, Restriction fragment length polymorphism.

[a]Study carried out on selected microscopy-positive samples.

[b]Longitudinal study. Infection rate calculated over 786 stool samples collected from 100 individuals.

[c]Initially reported as *Cryptosporidium* deer-like genotype.

*histolytica/dispar*, and *Blastocystis* sp. in apparently healthy schoolchildren from two districts in Lusaka Province, Zambia.

## 2. Materials and Methods

### 2.1 Ethics statement

This study was approved by the Levy Mwanawasa Medical University Ethics Review Committee (Ref. LMMU-REC000054/23) and the National Health Research Authority (NHRA0001/23/04/2023), Lusaka, Zambia. Prior to conducting the study, permission was sought from the Provincial and District Health and Education offices. Further, meetings were held with teachers, clinical staff, and community health workers in the respective study areas. Signed informed consents and assent were obtained from the parents/legal guardians of the participating schoolchildren and the children, respectively.

### 2.2. Study design and sampling area

A prospective, transversal epidemiological study was conducted to determine the presence of some of the most common diarrhoea-causing intestinal protists in schoolchildren aged 5–18 years attending primary schools in Chongwe (n = 2) and Kafue (n = 3) Districts in Lusaka Province, Zambia (Fig 1). The survey was conducted between May and August 2023. A

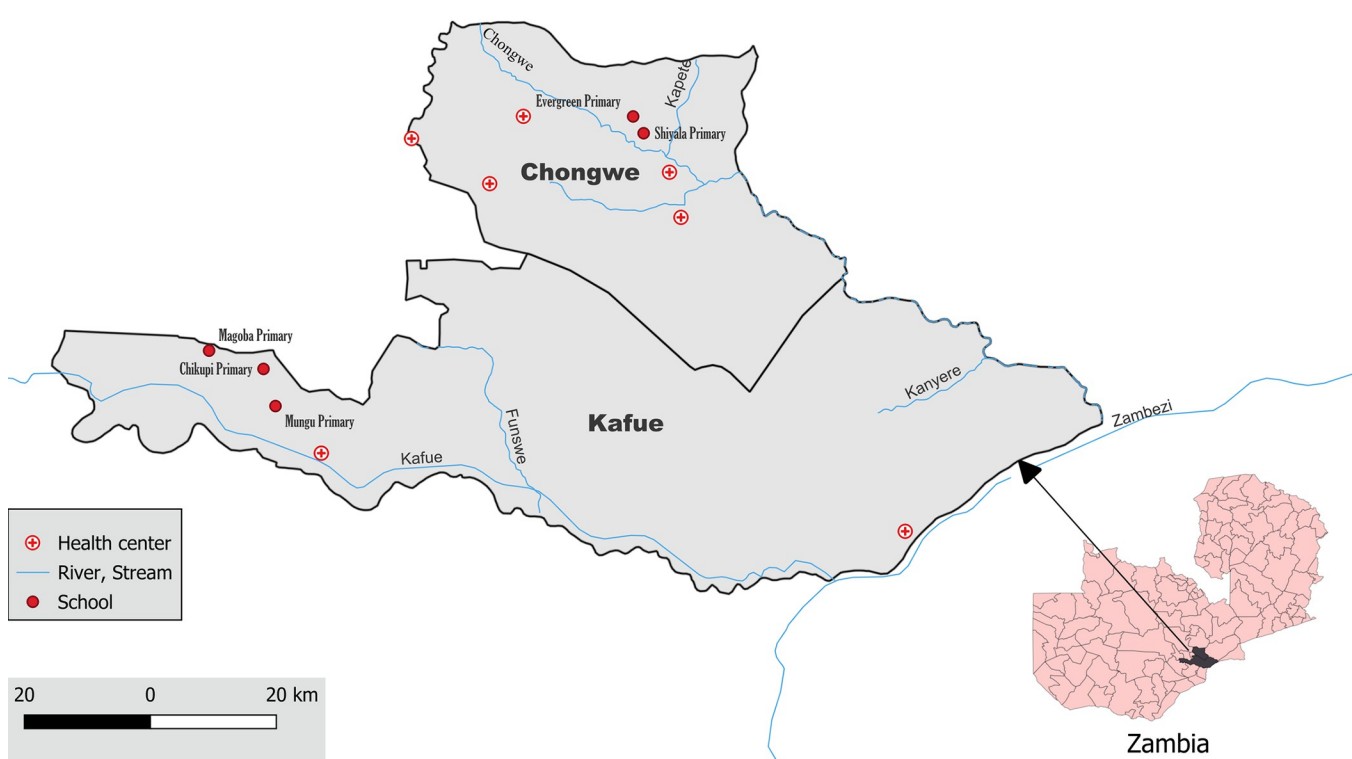

**Fig 1. Map of the Chongwe and Kafue Districts in Lusaka Province showing the location of the sampled primary schools.** The lower right corner of the Figure shows the geographical location of the Lusaka Province (coloured in black) within Zambia. The base layer of the map was obtained from https://gadm.org/maps.html. The GADM data is freely available for academic and non-commercial use and is compatible with the CC BY 4.0 license.

nonprobability convenience sampling strategy was followed, making the estimation of sample size unnecessary [44]. Variables potentially associated with higher odds of carrying enteric protists were also analysed. Chongwe District (population: 315,121 inhabitants), located about 35 km east of Lusaka, the capital of Zambia, is a semi-rural community with an important agricultural sector and small-scale livestock farming. Kafue District (population: 219,957 inhabitants), located south of Lusaka, is also a semi-rural community with agriculture and fishing as the main traditional occupations in the area.

We used molecular methods to detect and genotype (when possible) the pathogens under investigation. We recruited apparently healthy individuals of paediatric age because i) they are at higher risk of infection by intestinal protists due to exposure habits, ii) asymptomatic carriage of intestinal protists is insufficiently studied in sub-Saharan countries, and iii) sampling schoolchildren populations is easier to conduct from a logistic point of view, particularly in endemic areas such as Zambia.

### 2.3. Faecal sample and basic metadata collection

Faecal samples were collected from schoolchildren in Kafue (*n* = 158) and Chongwe (*n* = 109) districts. Participating schoolchildren were provided with uniquely labelled sampling containers (10 mL polystyrene plastic tube with spatula) to obtain individual stool samples. Collected samples were transported to the University Teaching Hospital (Lusaka) for storage at –40˚C until further processing.

A basic standardized epidemiological questionnaire was used to gather information on factors potentially associated with an increased likelihood of infection by intestinal protists.

Questions included: i) demographic features (age, gender, district of residence), ii) behavioural habits (consumption of unsafe water, contact with domestic animals), and iii) access to sanitary facilities. The questionnaire was completed by a member of our research staff through personal interviews with the children at sampling.

Only stool samples associated with completed questionnaires and signed informed consents were included in the survey.

## 2.4. DNA extraction and purification

Aliquots of the collected stool samples were shipped to the Parasitology Reference and Research Laboratory of the National Centre for Microbiology (Majadahonda, Spain) for DNA extraction and purification and PCR testing. Genomic DNA was isolated from about 200 mg of each faecal specimen by using the QIAamp DNA Stool Mini Kit (Qiagen, Hilden, Germany) according to the manufacturer's instructions, except that samples mixed with InhibitEX buffer were incubated for 10 min at 95°C. Extracted and purified DNA samples were eluted in 200 μL of PCR-grade water and kept at 4°C until further molecular analysis.

## 2.5. Molecular detection and characterization of *Giardia duodenalis*

*Giardia duodenalis* DNA was detected using a real-time PCR (qPCR) assay targeting the gene codifying the small subunit ribosomal RNA (*ssu* rRNA) of the parasite [45]. *Giardia*-positive isolates yielding cycle threshold ($C_T$) values ≤34 by qPCR were subsequently reassessed by a nested PCR to amplify a fragment of the *ssu* rRNA gene [46,47] to determine the molecular diversity of the parasite at the assemblage level. Samples that tested positive by *ssu*-PCR were re-amplified at the genes codifying the glutamate dehydrogenase (*gdh*), β-giardin (*bg*), and triose phosphate isomerase (*tpi*) proteins to determine the molecular diversity of the parasite at the sub-assemblage level. A semi-nested PCR was used to amplify a fragment of the *gdh* gene [48], and nested PCRs were used to amplify fragments of the *bg* and *tpi* genes, respectively [49–51].

## 2.6. Molecular detection of *Cryptosporidium* spp.

The presence of *Cryptosporidium* spp. was assessed using a nested-PCR protocol to amplify a 587-bp fragment of the *ssu* rRNA gene of the parasite [52]. A subtyping tool based on the amplification of a ∼850-bp partial sequence of the 60-kDa glycoprotein (*gp60*) gene was used to ascertain intra-species genetic diversity in samples that tested positive for *C. hominis* or *C. parvum* [53].

## 2.7. Molecular differential detection of *Entamoeba histolytica* and *Entamoeba dispar*

Detection and differential diagnosis between pathogenic *E. histolytica* and non-pathogenic *E. dispar* was carried out by a qPCR method targeting a 172-bp fragment of the *ssu* rRNA gene of the *E. histolytica*/*E. dispar* complex as described previously [54,55].

## 2.8. Molecular detection of *Blastocystis* sp.

*Blastocystis* sp. was identified by a single-round PCR protocol targeting a 600-bp of the *ssu* rRNA gene of the parasite [56].

## 2.9. PCR and gel electrophoresis standard procedures

All qPCR protocols described above were performed in a Corbett Rotor Gene 6000 qPCR system (Qiagen). Reaction mixes included 2× TaqMan Gene Expression Master Mix (Applied Biosytems, Foster City, CA). All the single-round, semi-nested, and nested PCR protocols described above were carried out in a 2720 Thermal Cycler (Applied Biosystems). Reaction mixes included 2.5 units of MyTAQ DNA polymerase (Bioline GmbH, Luckenwalde, Germany), and 5× MyTAQ Reaction Buffer containing 5 mM dNTPs and 15 mM MgCl$_2$. The specific DNA primer and probe sequences and PCR protocols used in the present study are detailed in S1 and S2 Tables, respectively. Laboratory-confirmed positive and negative DNA samples of human or animal origin for each parasitic species investigated were included as controls in each round of PCR.

PCR amplicons were visualized in 1.5–2% D5 agarose gels (Condalab, Madrid, Spain) stained with Pronasafe (Condalab) nucleic acid staining solution. A 100-bp DNA ladder (Boehringer Mannheim GmbH, Baden-Wurttemberg, Germany) was used for the sizing of obtained amplicons.

## 2.10. Sanger sequencing

Positive-PCR products were sequenced in both directions using internal primer sets (S1 Table) by capillary electrophoresis using the BigDye Terminator chemistry (Applied Biosystems) on an on ABI PRISM 3130 automated DNA sequencer. Obtained chromatograms were visually inspected for quality control and for detecting the presence of ambiguous (double peak) positions. Sequences obtained in this study were deposited in GenBank under accession numbers PQ185660–PQ185667 and PQ213645–PQ213681 (*Giardia duodenalis*), PQ191442–PQ191445 (*Cryptosporidium* spp.), and PQ200214–PQ200222 (*Blastocystis* sp.).

## 2.11. *Blastocystis* subtype identification using next-generation amplicon sequencing

A subset of *Blastocystis*-positive (confirmed by Sanger sequencing, *n* = 40) and *Blastocystis*-suspected (amplicons of the expected size but unconfirmed by Sanger sequencing, *n* = 5) DNA samples were shipped to the Environmental Microbial and Food Safety Laboratory, United States Department of Agriculture (Beltsville, Maryland, USA) for subsequent analysis including ST confirmation, identification of ST mixed infections, and investigation of intra-ST genetic diversity. A next-generation amplicon sequencing (NGS) strategy was conducted to identify *Blastocystis* subtypes as previously described [57]. Briefly, a PCR using the primer set ILMN_Blast505_532F/ILMN_Blast998_1017 was used to amplify a ca. 500-bp fragment of the *ssu* rRNA gene. These primers were identical to Blast505_532F/Blast998_1017R [58], except for containing Illumina overhang adapter sequences on the 5′ end. Final libraries were quantified by Qubit fluorometric quantitation (Invitrogen, Carlsbad, CA, USA) before normalisation. A final pooled library concentration of 8 pM with 20% PhiX control was sequenced using an Illumina MiSeq and a 600 cycle v3 kit (Illumina, San Diego, CA, USA). Paired-end reads were processed and analysed with an in-house pipeline as previously described [57]. The nucleotide sequences generated using NGS in this study were deposited in GenBank under the accession numbers PQ336781-PQ336832.

## 2.12. Statistical analyses

Statistical significance of the categorical variables included in this study was assessed using the Chi-square test or Fisher's exact test, depending on the sample size. A *P*-value of less than 0.05

was considered indicative of significant differences among the variables. For variables with more than one category showing statistically significant results, the Bonferroni correction was applied to reduce the risk of Type I error in multiple comparisons and the adjusted alpha level was set to 0.01667. Samples for which information for a given variable was unknown were not included in the analysis.

## 3. Results

A total of 256 individual stool samples were collected from children aged 5–18 years (median: 9.0; standard deviation: 2.1) from two districts (Chongwe: 43.4%, 111/256; Kafue: 56.6%, 145/256) in Lusaka Province, Zambia. The male/female ratio was 1.1. Children in the age group of 5–9 years accounted for more than half (56.0%, 136/243) of the surveyed paediatric population. Children frequently reported contact with domestic animals (85.1%, 206/242) including livestock (53.4%, 110/206), poultry (80.1%, 165/206), and dogs (8.7%, 18/206). Contact with two or more animal species was very common. Most of the children drank untreated water from wells (67.9%, 165/243) and had access to toileting including latrines (86.9%, 212/244) and flushable toilets (13.1%, 32/244) (see S3 Table).

### 3.1. Occurrence of intestinal protists

PCR-based prevalence rates of intestinal protists in the investigated paediatric Zambian populations are summarised in Table 2.

Overall, *Blastocystis* was the most frequent intestinal protist in the surveyed paediatric population [37.9%, 97/256; 95% Confidence Interval (95% CI): 31.9–44.1]. Other intestinal protists found included *G. duodenalis* (30.9%, 79/256; 95% CI: 25.3–36.90), *E. dispar* (13.3%, 34/256; 95% CI: 9.4–18.1), and *Cryptosporidium* spp. (4.3%, 11/256; 95% CI: 2.2–7.6). *Entamoeba histolytica* was not detected in any of the stool samples analysed.

Co-infections were detected in 53 children in six different combinations. *Blastocysts + G. duodenalis* was the most frequent (50.9%, 27/53), followed by *Blastocysts + G. duodenalis + E. dispar* (18.9%, 10/53), *Blastocystis + E. dispar* and *G. duodenalis + E. dispar* (11.3%, 6/53 each combination), *Blastocystis + Cryptosporidium* spp. (5.7%, 3/53), and *Cryptosporidium* spp. + *E. dispar* (1.9%, 1/53) (Fig 2).

### 3.2. Analysis of variables potentially associated with infections by intestinal protists

Children living in Kafue District were more likely to be positive for *G. duodenalis* (*P*-value: 0.001) and *Blastocystis* (*P*-value: 0.001), whereas for *Cryptosporidium* spp. positivity was significantly higher in children living in Chongwe (*P*-value: 0.01). None of the other variables, including age, sex, contact with animals, source of drinking water, and access to sanitary toilets, were positively associated with a higher likelihood of being positive for any of the intestinal protists investigated.

### 3.3. Molecular characterization of *Giardia duodenalis* isolates

All 79 faecal DNA samples with a positive result by qPCR yielded $C_T$ values ranging from 16.3 to 39.8 (median: 32.4; SD: 5.6). Of them, 59.5% (47/79) had $C_T$ values ≤34 and were reanalysed for genotyping purposes at the *ssu* rRNA locus (see subsection 2.5). We achieved successful amplifications in 31 samples. Sequence analysis allowed the identification of assemblages A (4/31), B (25/31), and A+B (2/31) (Table 3).

**Table 2. Prevalence of intestinal protist species in the surveyed paediatric population (*n* = 256) according to main sociodemographic variables.** Statistically significant values are bolded.

| Variable | No. | *Giardia duodenalis* | | | *Cryptosporidium* spp. | | | *Entamoeba dispar* | | | *Blastocystis* sp.[e] | | |
|---|---|---|---|---|---|---|---|---|---|---|---|---|---|
| | | Pos. (*n*) | % | *P*-value | Pos. (*n*) | % | *P*-value | Pos. (*n*) | % | *P*-value | Pos. (*n*) | % | *P*-value |
| District | | | | | | | | | | | | | |
| Chongwe | 111 | 15 | 13.5 | | 9 | 8.1 | | 14 | 12.6 | | 29 | 26.1 | |
| Kafue | 145 | 64 | 44.1 | **0.001** | 2 | 1.4 | **0.01** | 20 | 13.8 | 0.78 | 68 | 46.9 | **0.001** |
| Sex | | | | | | | | | | | | | |
| Male | 127 | 38 | 29.9 | | 5 | 3.9 | | 20 | 15.7 | | 49 | 38.6 | |
| Female | 118 | 34 | 28.8 | 0.84 | 6 | 5.1 | 0.70 | 14 | 11.9 | 0.38 | 42 | 35.6 | 0.63 |
| Unknown[a] | 11 | 7 | 63.6 | | 0 | 0.0 | | 0 | 0.0 | | 6 | 54.5 | |
| Age group (yrs.) | | | | | | | | | | | | | |
| 5–9 | 136 | 31 | 22.8 | | 10 | 7.4 | | 16 | 11.8 | | 53 | 39.0 | |
| 10–14 | 103 | 40 | 38.8 | 0.03[d] | 1 | 1.0 | 0.06[d] | 18 | 17.5 | 0.40 | 36 | 35.0 | 0.65 |
| 15–18 | 4 | 1 | 25.0 | 1[d] | 0 | 0.0 | 1[d] | 0 | 0.0 | | 2 | 50.0 | |
| Unknown[a] | 13 | 7 | 53.8 | | 0 | 0.0 | | 0 | 0.0 | | 6 | 46.2 | |
| Animal contact | | | | | | | | | | | | | |
| Yes[b] | 206 | 62 | 30.1 | | 11 | 5.3 | | 30 | 14.6 | | 80 | 38.8 | |
| No | 36 | 10 | 27.8 | 0.08 | 0 | 0.0 | 0.4 | 3 | 30.0 | 0.43 | 10 | 27.8 | 0.2 |
| Unknown[a] | 14 | 7 | 50.0 | | 0 | 0.0 | | 1 | 7.1 | | 7 | 50.0 | |
| Drinking water | | | | | | | | | | | | | |
| Surface[c] | 15 | 2 | 13.3 | | 0 | 0.0 | | 2 | 13.3 | | 4 | 26.7 | |
| Well | 165 | 57 | 34.5 | 0.15 | 7 | 4.2 | 0.76 | 21 | 12.7 | 0.63 | 67 | 40.6 | 0.31 |
| Tap | 63 | 12 | 19.0 | | 4 | 6.3 | | 11 | 17.5 | | 20 | 31.8 | |
| Unknown[a] | 13 | 8 | 61.5 | | 0 | 0.0 | | 0 | 0.0 | | 6 | 46.2 | |
| Toilet access | | | | | | | | | | | | | |
| Latrine | 212 | 66 | 31.1 | | 9 | 4.2 | | 29 | 13.7 | | 78 | 36.8 | |
| Flushable | 32 | 5 | 15.6 | 0.07 | 2 | 6.3 | 0.64 | 5 | 15.6 | 0.78 | 12 | 37.5 | 0.93 |
| Unknown[a] | 12 | 8 | 66.7 | | 0 | 0.0 | | 0 | 0.0 | | 7 | 58.3 | |

[a]Samples not included in the analysis.

[b]Including cattle (*n* = 43), sheep (*n* = 2), and goats (*n* = 85), and pigs (*n* = 19), dogs (*n* = 18), cats (*n* = 2), rabbits (*n* = 1), and ostriches (*n* = 1).

[c]Including streams and rivers.

[d]Bonferroni-adjusted *P*-values. The Bonferroni-adjusted significance was set to 0.01667.

[e]Based on samples confirmed by Sanger sequencing only.

Out of the four assemblage A sequences, two were identical to reference sequence M54878, and the other two differed from it by one single nucleotide polymorphism (SNP). Out of the 25 assemblage B sequences, 16 were identical to reference sequence AF199447, and the other nine differed from it by one SNP. Three out of four of these SNPs involved the presence of ambiguous (double peak) positions (S4 Table). All 31 samples with a positive result in *ssu*-PCR were subsequently analysed for sub-genotyping purposes at the *gdh*, *bg*, and *tpi* loci.

We successfully amplified at the *gdh* locus 67.7% (21/31) of the samples with a previous positive result in *ssu*-PCR. Sequence analyses confirmed the presence of sub-assemblages AII (23.8%, 5/21), BIII (4.8%, 1/21), BIV (47.6%, 10/21), and ambiguous BIII/BIV (23.8%, 5/21) (Table 3). All five AII sequences were identical to reference sequence L40510. The BIII sequence presented three SNPs when aligned with reference sequence AF069059. All 10 BIV sequences differed by 1–7 SNPs from reference sequence L40508 and only two were identical

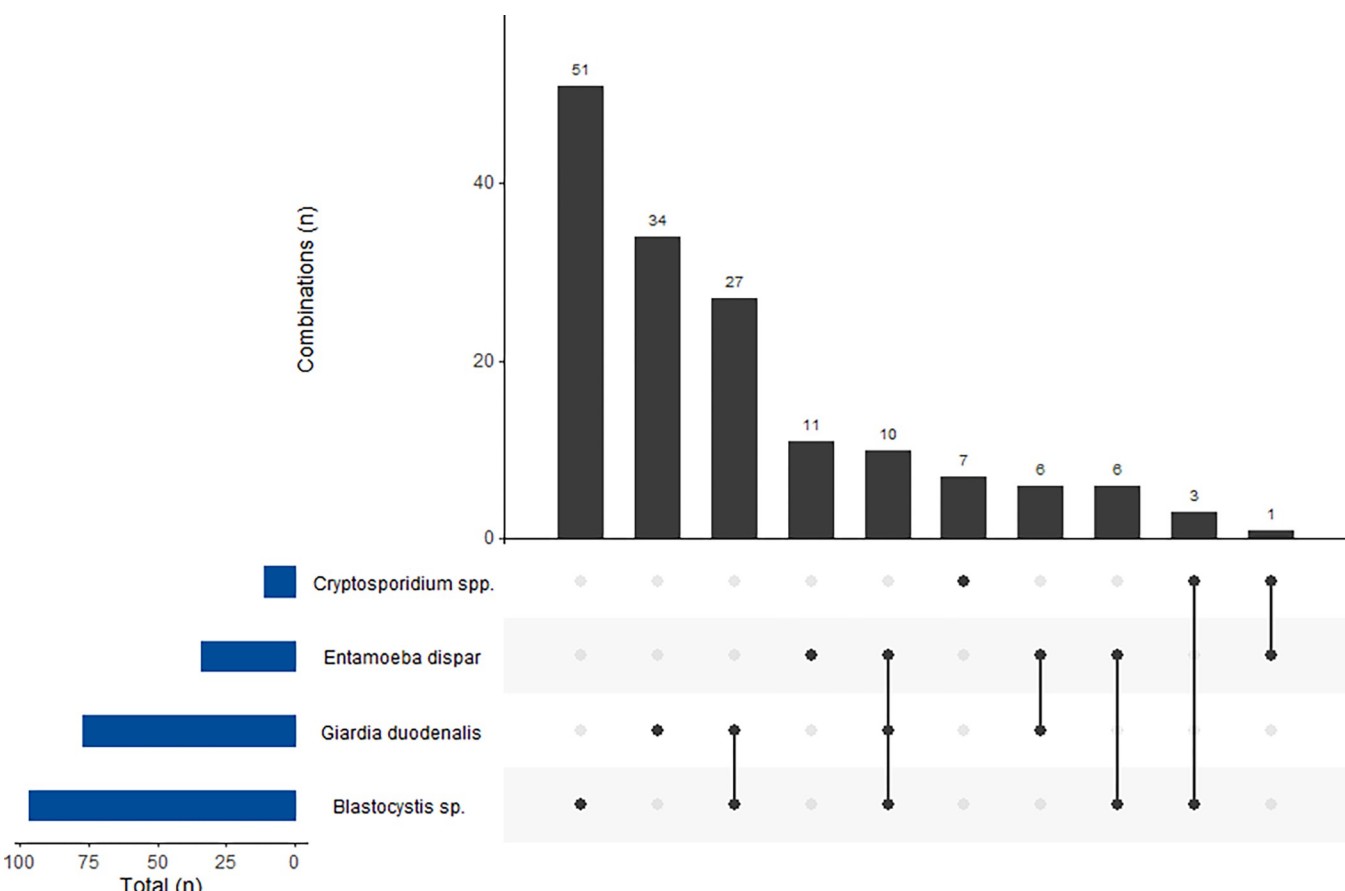

**Fig 2. Frequency of mono- and co-infections by intestinal protists identified in the present study.**

between them. A large genetic variability was observed among ambiguous BIII/BIV sequences, which varied by 5–12 SNPs from reference sequence L40508, most of them in the form of double peaks at chromatogram inspection (S4 Table).

We successfully amplified at the *bg* locus 54.8% (17/31) of the samples that previously tested positive by *ssu* rRNA-PCR. Sequence analyses allowed the identification of sub-assemblage AIII (23.5%, 4/17) and assemblage B (76.5%, 13/17) (Table 3). All four AIII sequences were identical to reference sequence AY072724, whereas the 13 assemblage B sequences varied by 1–6 SNPs from reference sequence AY072727 (S4 Table).

We successfully amplified at the *tpi* locus 45.2% (14/31) of the samples that previously tested positive by *ssu* rRNA-PCR. Sequence analyses allowed the identification of sub-assemblages AII (28.6%, 4/14), BIII and ambiguous BIII/BIV (35.7%, 5/14 each) (Table 3). Out of the four AII sequences, two were identical to reference sequence U57897, and the remaining two differed from it by a single SNP. All five BIII were distinct among them, differing by 1–7 SNPs from reference sequence AF069561. Similar to the *gdh* marker, we observed a large genetic diversity within BIII/BIV sequences, which differed by 7–16 SNPs from reference sequence AF069561 (S4 Table).

Considering all four loci used for genotyping/sub-genotyping purposes, assemblage B was the predominant genetic variant found (71.0%, 22/31), followed by co-infections by assemblages A+B (16.1%, 5/31), and assemblage A (12.9%, 4/31). Sub-assemblages AII (6.5%, 2/31),

**Table 3. Typing results of the 31 *Giardia duodenalis*-positive samples of paediatric origin successfully genotyped at any of the four loci (*ssu* rRNA, *gdh*, *bg*, *tpi*) investigated in the present survey.** The district of origin, age, and sex of the infected individuals are also shown.

| Sample ID | District | Sex | Age (yrs.) | *ssu* rRNA | *gdh* | *bg* | *tpi* | Assigned genotype |
|---|---|---|---|---|---|---|---|---|
| KMG46 | Kafue | Unknown | Unknown | B | – | – | – | B |
| KMG72 | Kafue | Unknown | Unknown | A | AII | AIII | AII | AII/AIII |
| KMG11 | Kafue | F | 11 | B | BIV | – | BIII | BIII/BIV |
| KMG37 | Kafue | M | 8 | B | BIV | B | BIII/BIV | BIII/BIV |
| KMG39 | Kafue | M | 13 | B | BIV | B | BIII/BIV | BIII/BIV |
| KMG40 | Kafue | F | 8 | B | BIV | B | BIII/BIV | BIII/BIV |
| KMG41 | Kafue | F | 8 | B | BIII/BIV | B | BIII | BIII/BIV |
| KMG43 | Kafue | F | 10 | B | BIII/BIV | B | BIII | BIII/BIV |
| KMG49 | Kafue | F | 13 | B | BIV | B | – | BIV |
| KMG69 | Kafue | Unknown | Unknown | B | BIV | – | – | BIV |
| KMG71 | Kafue | Unknown | Unknown | B | BIV | – | – | BIV |
| C03 | Kafue | F | 8 | B | – | – | – | B |
| C11 | Kafue | F | 11 | B | BIII/BIV | B | BIII/BIV | BIII/BIV |
| KCP15 | Kafue | M | 10 | B | – | – | – | B |
| KCP52 | Kafue | F | 12 | B | BIV | – | – | BIV |
| KCP55 | Kafue | M | 11 | B | – | – | – | B |
| KCP58 | Kafue | F | 12 | A | AII | – | AII | AII |
| KCP69 | Kafue | M | 11 | B | – | – | – | B |
| KCP70 | Kafue | F | 11 | B | BIV | – | BIII | BIII/BIV |
| KCP75 | Kafue | M | 12 | A | AII | – | – | AII |
| EG021 | Chongwe | F | 9 | B | – | – | – | B |
| EG024 | Chongwe | F | 8 | B | BIII/BIV | AIII | BIII | AIII+BIII/BIV |
| EG031 | Chongwe | F | 8 | A/B | AII | B | – | AII+B |
| EG101 | Chongwe | F | 9 | A/B | BIII | B | AII | AII+BIII |
| SC041 | Chongwe | F | 8 | B | – | B | BIII/BIV | BIII |
| KMU086 | Kafue | M | 7 | B | AII | AIII | – | AII/+AIII/B |
| KMU045 | Kafue | M | 7 | B | BIII/BIV | B | AII | AII+BIII/BIV |
| KMU052 | Kafue | M | 7 | A | – | AIII | – | AIII |
| KMU033 | Kafue | F | 7 | B | – | – | – | B |
| KMU018 | Kafue | F | 7 | B | – | B | – | B |
| KCP028 | Kafue | M | 9 | B | – | – | – | B |

*bg*: β-giardin; *gdh*: Glutamate dehydrogenase; *ssu* rRNA: Small subunit ribosomal RNA; *tpi*: Triose phosphate isomerase.

AIII (3.2%, 1/31), and AII/AIII (3.2%, 1/31) were identified within assemblage A, and BIV (12.9%, 4/31) and BIII/BIV (29.0%, 9/31) within assemblage B. A total of nine samples (29.0%, 9/31) were assigned to assemblage B with unknown sub-assemblage (Table 3).

### 3.4. Molecular characterization of *Cryptosporidium* spp. isolates

Sequence analyses of the 11 samples that tested positive for *Cryptosporidium* spp. at the *ssu* rRNA locus allowed the identification of *C. hominis* (18.2%, 2/11) and *C. parvum* (81.8%, 9/11). Both *C. hominis* sequences were identical to each other and to reference sequence AF108865. The nine *C. parvum* sequences differed by 4–6 SNPs with reference sequence AF112571. None of the samples positive for *C. hominis* or *C. parvum* could be amplified at the *gp60* locus.

### 3.5. Molecular characterization of *Blastocystis* sp. isolates

A total of 97 *Blastocystis*-positive samples were successfully genotyped in a preliminary Sanger sequence analysis revealing the presence of three subtypes including ST1 (35.1%, 34/97), ST2 (44.3%, 43/97), and ST3 (20.6%, 20/97).

A subset of samples ($n$ = 45) including 40 *Blastocystis*-positive from the above-mentioned 97 samples (confirmed by Sanger sequencing) and five *Blastocystis*-suspected (unconfirmed by Sanger sequencing) were reanalysed by NGS (Table 4).

Within the 45 samples analysed by NGS, ST1 was present in 51.1% (23/45), ST2 in 62.2% (28/45) and ST3 in 31.1% (14/45) (Table 4), as NGS analyses allowed the identification of all five samples in which Sanger sequencing did not provide definitive results. More than half of the initial *Blastocystis*-positive (by Sanger sequencing) samples (65%, 26/40) corresponded to mono-infections (ST1: 8; ST2: 14; ST3: 4). However, several co-infections with different STs were identified by NGS in samples initially considered as mono-infections by Sanger sequencing. These include ST1+ST2 ($n$ = 1) in a sample initially considered as ST1, ST1+ST2 ($n$ = 4), ST2+ST3 ($n$ = 3), and ST1+ST2+ST3 ($n$ = 3) in samples initially considered as ST2, and ST1+ST2 ($n$ = 1) and ST1+ST3 ($n$ = 2) in samples initially considered as ST3 (Table 4). As for the *Blastocystis*-suspected (unconfirmed by Sanger sequencing) samples, NGS analyses allowed the identification of both mono-infections (ST1: 1; ST2: 2) and co-infections (ST1+ST2: 1; ST1+ST3: 2) (Table 4).

Multiple unique sequence variants of ST1 and ST2 were frequently observed in individual samples (Table 4), but multiple variants of ST3 (up to two) were only detected in a single sample. In fact, up to three unique sequence variants were detected for ST1 and ST2 in three and two samples, respectively (Table 4). Thus, co-infection with multiple variants of ST3 appear to be less common than co-infection with multiple variants of ST1 and ST2. Overall, fifty-two unique sequence variants were detected among the three *Blastocystis* subtypes (ST1-ST3) (Table 5). ST2 had the highest intra-subtype variability, with 27 unique variants among 28 ST2-positive samples, followed by ST1, with 19 unique variants among the 23 ST1-positive samples (Tables 4 and 5). Subtype 3 displayed the least intra-subtype diversity, with only six unique variants among 14 ST3-positive samples. Unique sequence variants were relatively evenly distributed among samples, although several unique sequence variants were observed in multiple children (Table 5) indicating some sequence variants may be more common in this population.

## 4. Discussion

PCR-based prevalence data and molecular information on the frequency of species/genotypes of diarrhoea-causing intestinal protists are limited in sub-Saharan African countries. Strengths of this study include the use of i) highly sensitive and specific molecular methods for the detection and differential diagnosis of protists of public health significance, ii) Sanger and next-generation sequencing methods for genotyping and sub-genotyping purposes and (in the case of *Blastocystis*) identification of mixed STs within a sample, and iii) a relatively large panel of apparently healthy schoolchildren to investigate the potential role of asymptomatic individuals as unnoticed spreaders of intestinal protists.

This is the first study conducted in Zambia in which PCR was used as a screening method for the detection of *G. duodenalis*, *Cryptosporidium* spp., *E. histolytica*, and *Blastocystis* sp. *Giardia duodenalis* was identified in 31% of the investigated children, which is the largest figure documented in any Zambian human populations so far. Previously, prevalence rates of 2–10% were reported by conventional microscopy [24,25,28,41] and of 19–28% by IFA [26,32,33]. Most of these studies were conducted in districts of the Lusaka province and most

**Table 4. Comparison of the performance of Sanger and next-generation sequencing methods for the identification of *Blastocystis* subtypes in a subset of samples (*n* = 45) and intra-subtype genetic diversity observed in individual samples.**

| Sample ID | Sanger sequencing | Next-generation sequencing | |
|---|---|---|---|
| | ST | ST(s) | ST variants (%) |
| KMG63 | ST2 | ST2* | ST2 (77)/ST2 (23) |
| KMG64 | ST1 | ST1* | ST1 (51)/ST1 (49) |
| KMG65 | ST1 | ST1 | ST1 (100) |
| KMG66 | ST1 | ST1 | ST1 (100) |
| KMG71 | ST2 | ST2* | ST2 (56)/ST2 (44) |
| KMG72 | ST1 | ST1* | ST1 (75)/ST1 (24)/ST1 (1) |
| KMG11 | ST1 | ST1* | ST1 (97)/ST1 (3) |
| KMG12 | ST2 | ST2*+ST3 | ST2 (45)/ST2 (43)/ST3 (10)/ST2 (2) |
| KMG18 | Untypable | ST1*+ST3 | ST3 (74)/ST1 (20)/ST1 (6)/ST1 (<1) |
| KMG19 | ST3 | ST1+ST3 | ST3 (82)/ST1 (18) |
| KMG21 | ST3 | ST1*+ST2* | ST2 (58)/ST2 (32)/ST1 (6)/ST1 (4) |
| KMG36 | ST2 | ST2+ST3* | ST3 (93)/ST2 (7)/ST3 (<1) |
| KMG39 | Untypable | ST1*+ST3 | ST3 (72)/ST1 (28)/ST1 (<1) |
| KMG40 | ST3 | ST1*+ST3 | ST3 (98)/ST1 (2)/ST1 (<1) |
| KMG41 | ST2 | ST1+ST2*+ST3 | ST2 (83)/ST3 (7)/ST2 (7)/ST1 (3) |
| KMG43 | ST2 | ST1+ST2*+ST3 | ST2 (82)/ST3 (9)/ST2 (6)/ST1 (3) |
| KMG49 | ST1 | ST1+ST2 | ST1 (98)/ST2 (2) |
| KMG71 | ST2 | ST2* | ST2 (61)/ST2 (31)/ST2 (8) |
| C03 | ST2 | ST1+ST2*+ST3 | ST2 (57)/ST2 (24)/ST3 (16)/ST1 (3) |
| KCP03 | ST1 | ST1* | ST1 (75)/ST1 (23)/ST1 (2) |
| KCP08 | ST2 | ST1+ST2* | ST2 (57)/ST2 (43)/ST1 (<1) |
| KCP13 | ST3 | ST3 | ST3 (100) |
| KCP15 | ST2 | ST1+ST2* | ST2 (60)/ST2 (38)/ST1 (2) |
| KCP36 | ST2 | ST2* | ST2 (82)/ST2 (18) |
| KCP41 | ST2 | ST2* | ST2 (74)/ST2 (26) |
| KCP51 | ST2 | ST2* | ST2 (77)/ST2 (23) |
| KCP53 | ST2 | ST2* | ST2 (76)/ST2 (24) |
| KCP54 | ST2 | ST1*+ST2* | ST2 (87)/ST2 (6)/ST1 (5)/ST1 (2) ST2 (93)/ST2 (7) |
| KCP55 | ST2 | ST2* | ST2 (75)/ST2 (25) |
| KCP58 | Untypable | ST1 | ST1 (100) |
| KCP60 | Untypable | ST2* | ST2 (58)/ST2 (42) |
| KCP62 | ST2 | ST2* | ST2 (76)/ST2 (24) |
| KCP69 | ST1 | ST1 | ST1 (100) |
| KCP70 | ST3 | ST3 | ST3 (100) |
| KCP73 | ST2 | ST2* | ST2 (99)/ST2 (1) |
| KCP83 | ST3 | ST2* | ST2 (71)/ST2 (29) |
| KCP88 | ST2 | ST2* | ST2 (92)/ST2 (8) |
| KCP72 | ST2 | ST1+ST2* | ST1 (94)/ST2 (6)/ST2 (<1) |
| KCP71 | ST2 | ST3 | ST3 (100) |
| KCP75 | ST2 | ST2* | ST2 (83)/ST2 (17) |
| KCP84 | ST2 | ST2+ST3 | ST3 (100)/ST2 (<1) |
| KCP85 | Untypable | ST1*+ST2* | ST1 (50)/ST2 (41)/ST2 (6)/ST1 (3) ST2 (88)/ST2 (12) |
| KCP86 | ST1 | ST1* | ST1 (92)/ST1 (8) |

*(Continued)*

**Table 4.** (Continued)

| Sample ID | Sanger sequencing | Next-generation sequencing | |
|-----------|-------------------|-----------|------|
| | ST | ST(s) | ST variants (%) |
| KCP94 | ST2 | ST2* | ST2 (45)/ST2 (43)/ST2 (12) |
| KCP96 | ST3 | ST3 | ST3 (100) |

*Sample showing any intra-subtype genetic diversity.

targeted paediatric populations (Table 1). Overall, these data indicate that PCR-based methods provide better sensitivity for the detection of giardiasis, even when samples screened are from asymptomatic subjects in which lower number of parasites could be expected.

*Cryptosporidium* infections were detected in 4% of the investigated children. That number is within the range of previously reported studies in Zambia, where cryptosporidiosis cases have been reported at highly variable prevalence rates ranging from of 2% to 28% by conventional microscopy in paediatric, adult, and HIV-positive patient populations [23–26,28–30,36]. A prevalence rate of 6% has been identified by ELISA in farm workers [31] and of 28–31% by IFA in children [32,33]. As in the case of *G. duodenalis*, all studies investigating the presence of *Cryptosporidium* spp. were conducted in Lusaka Province. The high discrepancy rates on *Cryptosporidium* prevalence among those studies suggest the need of conducting comparative performance analyses of the detection test used to determine their true accuracy and robustness.

Remarkably, we did not detect the presence of *E. histolytica* in any of the 256 faecal samples analysed by PCR. Instead, we found *E. dispar* in 13.3% of them. This finding highlights the relevance of molecular methods for the differential diagnosis of pathogenic *E. histolytica* and the non-pathogenic members of the *Entamoeba* complex, from which *E. dispar* is the most common species. It should be also noted that conventional methods (particularly microscopy examination) are not suitable for such purpose and lead to the overdiagnosis of *E. histolytica* [59]. In Zambia, *E. histolytica* has been reported previously in paediatric and adult HIV-positive patients by conventional microscopy [23], but this result should be interpreted with caution.

In the present study, *Blastocystis* sp. was the most prevalent (34%) intestinal protist found in the surveyed paediatric population. This carriage rate was in the upper range of those (2–54%) documented previously by conventional microscopy in paediatric, adult, and HIV-positive populations in Zambia [23–26]. It should be noted that *Blastocystis* is a highly polymorphic microorganism with four major (vacuolar, cyst, granular, and amoeboid) morphologic forms, a fact that impairs the identification of the protist by conventional microscopy.

Within *G. duodenalis* infections, assemblage B was more prevalent than assemblage A (71% *vs.* 13%) in the studied paediatric population. This result was in line with those reported previously by our research group using the same genotyping scheme in human populations from other sub-Saharan countries including Angola (64% *vs.* 36%) [60], Ethiopia (82% *vs.* 18%) [61], Mozambique (88–90% *vs.* 7–10%) [62,63], and Nigeria (68% *vs.* 28%) [64]. In contrast, assemblage A seems more prevalent in countries such as Egypt, Central Africa Republic, and Uganda [15], suggesting that assemblage frequency patterns could be geographically restricted and reflect differences in source of infection or transmission pathways. Interestingly, we also found a high proportion of mixed A+B infections (16%), which could be indicative of high infection pressures and elevated rates of re-infection. A large molecular variability was observed within assemblage B sequences, showing up to 12 SNPs at the *gdh* marker and up to 16 SNPs at the *tpi* marker compared with reference sequences. The fact that a large proportion

**Table 5.  Information on *Blastocystis* intra-subtype variants obtained by next generation amplicon sequencing.**

| Subtype | No. of unique ST variants | No. of samples containing unique ST variants | GenBank accession number |
|---------|---------------------------|----------------------------------------------|--------------------------|
| ST1 | 19 | 3 | PQ336784 |
|  |  | 8 | PQ336785 |
|  |  | 1 | PQ336789 |
|  |  | 1 | PQ336790 |
|  |  | 1 | PQ336795 |
|  |  | 4 | PQ336797 |
|  |  | 3 | PQ336798 |
|  |  | 3 | PQ336802 |
|  |  | 1 | PQ336811 |
|  |  | 1 | PQ336814 |
|  |  | 1 | PQ336816 |
|  |  | 1 | PQ336821 |
|  |  | 1 | PQ336823 |
|  |  | 2 | PQ336824 |
|  |  | 1 | PQ336825 |
|  |  | 2 | PQ336826 |
|  |  | 1 | PQ336829 |
|  |  | 1 | PQ336830 |
|  |  | 1 | PQ336831 |
| ST2 | 27 | 4 | PQ336782 |
|  |  | 7 | PQ336783 |
|  |  | 5 | PQ336787 |
|  |  | 3 | PQ336788 |
|  |  | 2 | PQ336791 |
|  |  | 2 | PQ336792 |
|  |  | 2 | PQ336793 |
|  |  | 5 | PQ336794 |
|  |  | 1 | PQ336796 |
|  |  | 2 | PQ336800 |
|  |  | 2 | PQ336801 |
|  |  | 1 | PQ336803 |
|  |  | 1 | PQ336804 |
|  |  | 1 | PQ336805 |
|  |  | 1 | PQ336806 |
|  |  | 3 | PQ336807 |
|  |  | 1 | PQ336808 |
|  |  | 1 | PQ336809 |
|  |  | 1 | PQ336810 |
|  |  | 4 | PQ336812 |
|  |  | 1 | PQ336813 |
|  |  | 1 | PQ336815 |
|  |  | 1 | PQ336817 |
|  |  | 1 | PQ336818 |
|  |  | 1 | PQ336822 |
|  |  | 1 | PQ336827 |
|  |  | 1 | PQ336832 |

(*Continued*)

**Table 5.** (Continued)

| Subtype | No. of unique ST variants | No. of samples containing unique ST variants | GenBank accession number |
|---------|---------------------------|-----------------------------------------------|--------------------------|
| **ST3** | 6 | 7 | PQ336781 |
|  |  | 3 | PQ336786 |
|  |  | 1 | PQ336799 |
|  |  | 2 | PQ336819 |
|  |  | 1 | PQ336820 |
|  |  | 1 | PQ336828 |

(24–36%) of the assemblage B sequences generated in this study corresponded to inconsistent BIII/BIV results (most of them associated with ambiguous nucleotide positions in the form of double peaks) evidenced the resolution limitations of the *gdh*, *bg*, and *tpi* loci, highlighting the need of identifying novel markers for accurate and robust molecular typing [65]. Absence of animal-adapted assemblages C-F seem to suggest that most of the *G. duodenalis* infections detected here are of anthropogenic nature.

We detected *C. parvum* more frequently than *C. hominis* (82% *vs.* 18%, respectively) in our paediatric population. This proportion was very similar to that (80% *vs.* 20%) identified in farm workers in Lusaka Province [31]. In contrast, *C. hominis* was the predominant species found in children in urban settings from different Zambian regions (88%) [36] and in patients in Lusaka (59%) [34]. In the latter survey, sporadic cases of human infections by *C. felis* and *C. meleagridis* were also described. Taken together, these findings suggest that individuals living in rural areas and with increased contact with livestock were more likely to harbour zoonotic *Cryptosporidium* infections, whereas those living in urban areas were more likely to harbour anthroponotic infections. Of note, *C. parvum* was also the most prevalent *Cryptosporidium* species found in cattle in Zambia [31,38]. Most reports of *Cryptosporidium* in livestock in Zambia were conducted in cattle, however, small ruminants are also in close contact with humans in rural communities potentially being a source of *Cryptosporidium* infections. Presence of *Cryptosporidium* spp. has been reported in sheep and goats, but unfortunately, there is no information of species present in those animals as no molecular data is currently available [43]. Remarkably, we were unable to genotype any of our *C. hominis*- or *C. parvum*-positive samples at the *gp60* gene, suggesting that the parasite burden in the infected children was probably low. This fact precluded us to unambiguously confirm the zoonotic nature of our *C. parvum* infections, as out of the three dominant *C. parvum gp60* subtype families in humans IIa and IId are zoonotic, whereas IIc appears to be almost exclusively anthroponotic. These findings should be confirmed and expanded in future molecular-based surveys targeting human and livestock (cattle and small ruminants) populations from other geographical areas of the country.

In agreement with previous studies conducted in sub-Saharan countries, we observed a limited *Blastocystis* subtype diversity within the surveyed paediatric population, where only ST1-ST3 were identified. These were also the most prevalent subtypes found in individuals with and without HIV infection in Ghana [66], in schoolchildren with and without gastrointestinal manifestations in Mozambique and Senegal (in addition to ST4) [63,67], in children in Nigeria and patients in Tanzania (in addition to ST7) [68,69], and in patients in Zambia (in addition to ST6) [27]. Notably, we confirmed the presence of multiple subtypes (ST1-ST3 in four different combinations) within a sample in 35.4% (17/48) of the *Blastocystis*-positive faecal samples examined by NGS. This technology has the advantage of allowing the identification of mixed STs, including underrepresented STs (low proportion) that are missed by Sanger

sequencing [70]. Our co-colonization rate was higher than those reported in previous epidemiological studies conducted in Colombia (28.2%, 20/71 involving ST1-ST5; 27.6%, 8/29, involving ST1-ST3) [71,72], the Czech Republic (7.2%, 6/83 involving ST1-ST3 and ST7) [73], Mexico (13.7%, 17/124 involving ST1-ST3) [74], and Turkey (20.6%, 14/68 involving ST1-ST3) [75], suggesting that *Blastocystis* is highly endemic in Zambia. The identification of only ST1-ST3, with the absence of other subtypes that although reported in humans are more frequently identified in animals (such as ST5, ST6, or ST7) may indicate that a large (but unknown) proportion of the *Blastocystis* infections identified in the present study may be anthroponotic in nature. It has been demonstrated that NGS is appropriate to investigate intra-subtype diversity within individual samples. *Blastocystis* subtype designations contain a fairly large amount of genetic diversity, and STs can vary by up to 4% within individual subtypes. Thus, investigating the degree of intra-subtype variability within populations and within individual hosts can assist in understanding the epidemiology of *Blastocystis*. Intra-subtype diversity was strikingly high in the present study, with 52 unique genetic variants observed in the 45 *Blastocystis*-positive samples sequenced using NGS. Of the three STs identified in this study, ST1 and ST2 had higher diversity than ST3. This lower level of intra-subtype diversity in ST3 and higher diversity in ST1 and ST2 have been previously reported in human populations of Mexico and Colombia [71,74] and support the idea that ST3 may have a more restricted source of transmission in human populations while sources of transmission for ST1 and ST2 may be more variable.

Schoolchildren living in Kafue District were significantly more likely to carry *Giardia* and *Blastocystis* than their counterparts living in Chongwe District. This discrepancy is difficult to explain considering that both are semi-rural communities with similar sociodemographic conditions. Local differences in access to safe drinking water, sanitary infrastructures, or exposure to zoonotic infection sources may explain, at least partially, the differences found. Indeed, *C. parvum* infections were significantly more frequent in schoolchildren living in Chongwe District, a region where livestock raising including cattle and small ruminants is an important economic activity but not in Kafue District. Interestingly, children aged 10–14 years were identified as the age group more susceptible to *Giardia* and *Cryptosporidium* infections. Findings from large prospective longitudinal cohort studies conducted in sub-Saharan African countries has shown that these intestinal protists primarily affect young children under five years of age as consequence of poor personal hygiene and immature immune systems [1,76,77]. Unfortunately, this age group was unavailable in this study, and we were unable to corroborate the findings reported in those previous studies.

This study is hampered by some limitations that should be considered when interpreting some of the results obtained and the conclusions reached. First, the survey was limited to paediatric populations within two districts in the Lusaka Province and the prevalence and molecular data obtained might not be representative of the whole Zambian scenario. Second, this is a transversal survey, so we were unable to follow up infections in individual children or to evaluate any potential seasonal variation of the investigated pathogens. Third, and as the participation in the study was voluntary, it is possible that families/children perceived as at higher risk of infection were more likely to participate. The distorting effect of this behaviour would have been more apparent in sampling areas with lower participation rates. And fourth, PCR resolution (i.e., *G. duodenalis* genotyping) or sensitivity (i.e., *Blastocystis* detection) issues very likely have negatively impacted the accuracy and robustness of some of the data presented here. This is why, in the case of *Blastocystis* sp., we adopted a conservative approach and considered as positive only Sanger-confirmed samples despite knowing that the reported prevalence rate was an underestimation of the true one.

In conclusion, we used molecular (PCR and Sanger/NGS sequencing) methods to provide updated information on the occurrence and molecular diversity of diarrhoea-causing protist parasites in an apparently healthy paediatric population in Lusaka Province, Zambia. Well-stablished pathogens *G. duodenalis* and *Cryptosporidium* spp. were identified at moderate-to-high infection rates in healthy children, suggesting that asymptomatic carriers may play a significant role in the spreading of intestinal pathogens. Human data presented here should be confirmed in other Zambian geographical regions and extended to domestic animal reservoirs and environmental (water) samples.

## Supporting information

**S1 Table. Oligonucleotides used for the molecular identification and/or characterization of the intestinal protists investigated in the present study.**
(DOCX)

**S2 Table. PCR cycling conditions used for the molecular identification and/or characterization of the intestinal protists investigated in the present study.**
(DOCX)

**S3 Table. Full dataset showing the sociodemographic, epidemiological, diagnostic, and molecular data generated in the present study.**
(XLSX)

**S4 Table. Frequency and molecular diversity of *G. duodenalis* identified at the *ssu* RNA, *gdh*, *bg*, and *tpi* loci in the paediatric population under study.** Lusaka Province, Zambia (2023). GenBank accession numbers are provided.
(DOCX)

**S5 Table. Frequency and molecular diversity of *Cryptosporidium* spp. identified at the *ssu* RNA locus in the paediatric population under study.** Lusaka Province, Zambia (2023). GenBank accession numbers are provided
(DOCX)

## Author Contributions

**Conceptualization:** Mable Mutengo, Kyoko Hayashida, James Mwansa, Javier Sotillo, David Carmena.

**Data curation:** Namwiinga R. Mulunda, Freeman W. Chabala, Stanley Chinyanta.

**Formal analysis:** Michaela Kaduková, Alejandro Dashti, Sergio Sánchez, Pamela C. Köster, Jenny G. Maloney, Mónica Santín.

**Funding acquisition:** Mable Mutengo, Mónica Santín, Javier Sotillo, David Carmena.

**Investigation:** Mable Mutengo, Michaela Kaduková, Alejandro Dashti, Lourdes Castro, Pamela C. Köster, David González-Barrio, Jenny G. Maloney.

**Methodology:** Mable Mutengo, Freeman W. Chabala.

**Project administration:** Mable Mutengo, Kelly Chisanga, Javier Sotillo, David Carmena.

**Resources:** Mable Mutengo, Mónica Santín, Javier Sotillo, David Carmena.

**Supervision:** Mable Mutengo, James Mwansa, Javier Sotillo, David Carmena.

**Validation:** Mable Mutengo, Sergio Sánchez, Mónica Santín, Javier Sotillo, David Carmena.

**Writing – original draft:** Mable Mutengo, Sergio Sánchez, Mónica Santín, Javier Sotillo, David Carmena.

**Writing – review & editing:** Mable Mutengo, Michaela Kaduková, Namwiinga R. Mulunda, Freeman W. Chabala, Alejandro Dashti, Kyoko Hayashida, Stanley Chinyanta, Kelly Chisanga, Lourdes Castro, Sergio Sánchez, James Mwansa, Pamela C. Köster, David González-Barrio, Jenny G. Maloney, Mónica Santín, Javier Sotillo, David Carmena.

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
