## [Decision Letter · Decision Letter 0]

31 Oct 2024

PNTD-D-24-01392Asymptomatic carriage of intestinal protists is common in children in Lusaka Province, ZambiaPLOS Neglected Tropical Diseases Dear Dr. Carmena, Thank you for submitting your manuscript to PLOS Neglected Tropical Diseases. After careful consideration, we feel that it has merit but does not fully meet PLOS Neglected Tropical Diseases's publication criteria as it currently stands. Therefore, we invite you to submit a revised version of the manuscript that addresses the points raised during the review process. Please submit your revised manuscript within 60 days Dec 30 2024 11:59PM. If you will need more time than this to complete your revisions, please reply to this message or contact the journal office at plosntds@plos.org. Please include the following items when submitting your revised manuscript:*
A rebuttal letter that responds to each point raised by the editor and reviewer(s). You should upload this letter as a separate file labeled 'Response to Reviewers'. This file does not need to include responses to any formatting updates and technical items listed in the 'Journal Requirements' section below.*
A marked-up copy of your manuscript that highlights changes made to the original version. You should upload this as a separate file labeled 'Revised Manuscript with Track Changes'.*
An unmarked version of your revised paper without tracked changes. You should upload this as a separate file labeled 'Manuscript'. If you would like to make changes to your financial disclosure, competing interests statement, or data availability statement, please make these updates within the submission form at the time of resubmission. Guidelines for resubmitting your figure files are available below the reviewer comments at the end of this letter. We look forward to receiving your revised manuscript. Kind regards, Luisa MagalhãesAcademic EditorPLOS Neglected Tropical Diseases Hira NakhasiSection EditorPLOS Neglected Tropical Diseases

Shaden Kamhawi

co-Editor-in-Chief

Paul Brindley

co-Editor-in-Chief

 **Journal Requirements:** **Additional Editor Comments (if provided):****Reviewers' Comments:** Reviewer's Responses to Questions

**Key Review Criteria Required for Acceptance?**

**Methods**

-Are the objectives of the study clearly articulated with a clear testable hypothesis stated?

-Is the study design appropriate to address the stated objectives?

-Is the population clearly described and appropriate for the hypothesis being tested?

-Is the sample size sufficient to ensure adequate power to address the hypothesis being tested?

-Were correct statistical analysis used to support conclusions?

-Are there concerns about ethical or regulatory requirements being met?

Reviewer #1: The objectives of the study were stated clearly, the study design is appropriate to address the objective. Study population, sample size and statistical analysis were performed correctly. Appropriate ethical and regulatory approvals were obtained. However, there are a few points that can help to improve the quality of the manuscript:

1.Was there any sample size calculation? I would expect to have this explanation, particularly given that this was a prospective, transversal epidemiological study (cross-sectional?)

2.How were participants for the study selected? Were there any specific inclusion and exclusion criteria? This is relevant, especially since risk factors are evaluated

3.Authors should indicate the timeframe for the study, i. e. when was the sample collection and data collection conducted in Zambia?

4.The authors indicate that the samples were stored for some time in the University Teaching Hospital (Lusaka) for some time, but where were the samples finally processed? This is only specified unambiguously for Blastocystis-positive and Blastocystis-suspected samples which were further analysed at the US Department of Agriculture.

5.For statistical analysis of epidemiological data, the authors presented p-values in Table 2. What statistical tests were applied? I assume it was chi-square or Fisher’s exact test, if that is the case, was a method to correct for multiple testing (Bonferroni) applied when more that two categories in a variable was compared? Examples: age group (5-9 years, 10-14 years, 15-18 years and unknown); drinking water. And for those variables with the category Unknown, how was it treated? Is it included in the statistical test or is it excluded?

Reviewer #2: acceptable

**Results**

-Does the analysis presented match the analysis plan?

-Are the results clearly and completely presented?

-Are the figures (Tables, Images) of sufficient quality for clarity?

Reviewer #1: The results are presented clearly and all relevant data is either included in the manuscript body, supplementary files and appropriate data repositories with unique identifiers. A few topics for consideration that may improve the quality of the manuscript:

6.In the “3.2. Analysis of risk factors potentially associated with infections by intestinal protists” sub-section, authors state that “Giardia duodenalis and Cryptosporidium infections were significantly more frequent in children belonging to the 10–14 years age group (P-values: 0.01 and 0.02, respectively)”, however, the data on Table 2, shows that the Cryptosporidium was more frequent in the 5-9 years age group, with 10 of the 11 infected participants. In addition, this would be in line with what has been documented by the GEMS study, where it was much more common in younger children (<2 years).

7.The authors present the frequency of the protists according to different variables (very often referring to them as risk factors). Both in the Results and the Discussion they state that children with a certain specific characteristic “… were significantly more likely …” to be infected/carry a given protist. However, since they only presented a comparison of frequencies between groups and not the results of association analysis (using, for example, logistic regression or other methods) the wording should be revised carefully, as it can be misleading. I understand that association analysis might be beyond the scope of this work.

Reviewer #2: acceptable

**Conclusions**

-Are the conclusions supported by the data presented?

-Are the limitations of analysis clearly described?

-Do the authors discuss how these data can be helpful to advance our understanding of the topic under study?

-Is public health relevance addressed?

Reviewer #1: The conclusions presented by the authors correspond to the results and data in the manuscript. Authors present the limitations of their study clearly and how this data advances the topic under study.

Reviewer #2: acceptable

**Editorial and Data Presentation Modifications?**

Reviewer #1: (No Response)

Reviewer #2: minor revision

**Summary and General Comments**

Reviewer #1: The manuscript presents data on a relevant topic and is an attempt to fill a gap in the literature for a topic that has largely been neglected in public health research, particularly in sub-Saharan Africa. Authors have used not only PCR and Sanger sequencing to perform the typical detection and basic genotyping of intestinal protists, but have also used a new approach (NGS amplicon sequencing) to explore and uncover mixed infections by Blastocystis. The manuscript is well written, however it can benefit from some minor adjustments and corrections, as suggested in the points that I raise below.

1.Line numbering: why does it only start on page 7?

2.Check spelling in the entire document and correct typos. E. g: in the abstract it reads “We used PCR for the initial screening of samples for the presence of intestinal protest species”, when it should be “protist”.

Reviewer #2: This study evaluated the asymptomatic intestinal protozoan infection in Zambia. High-accuracy molecular diagnostic approaches, including specific PCR and next-generation sequencing, were first utilized in this survey. The findings provided the prevalence of Cryptosboridium sp., Giardia duodenalis, Entamoeba dispar, and Blastocystis sp. in school-aged healthy individuals in Chongwe (158) and Kafue (109) districts in Lusaka Province. In addition, detailed information on subtypes and genetic variations was also documented. This investigation highlights the importance of asymptomatic carriers of intestinal protists in public health in the sub-Sahara area.

Generally, the manuscript seems almost acceptable in the current version; however, some issues should be addressed before further processing.

1.Lines 33 to 38:

The samples were collected from Kafue and Chongwe districts in Lusaka Province, Zambia. The epidemiology investigation provides valuable information for African countries and regions with similar geographical characteristics. Therefore, a map containing the geographical, hydrological, and livestock-keeping facilities in Lusaka Province, especially the Kafue and Chongwe districts, is recommended to be provided.

2.Page 6 to lines 13:

It is appreciated that the authors organized a list to emphasize the previous epidemiological surveys and the utilities of relatively low-sensitivity methods in Zambia (Table 1). However, the table, if possible, should be introduced/ discussed more or moved as a supplementary file.

3.Lines 169 to 173:

The co-infection of individuals with two or more protists was mentioned. However, further analysis was expected, such as the origin of samples, the livestock contact, etc. The discussion section should also include the possible speculation of the co-infection with different protists.

4.Lines 187 to 191:

The authors mentioned that “all 79 faecal DNA samples with a positive result …”, but only 31 could be amplified and typed. The detailed detection strategy was stated in the method section (from line 70). However, for clarity and to make the audience easy to understand, the screening process needed to be introduced in the result section, too.

5.Lines 244 to 249:

The authors stated that “A total of 97 Blastocystis-positive samples were successfully genotyped”, and the proportion of subtypes was revealed. It was unknown why the following paragraph started with “a subset of samples (n=45) …”. What were the missing samples out of the subset 45? Again, although the method section was stated, the origin of the subset should also be mentioned logically in the result section.

6.Lines 268 to 276:

The whole paragraph about intra-subtype variants, coupled with Table 5, is not easy to understand.

Table 5 showed the subtype, No. of unique ST variants, No. of samples containing variants, and GenBank accession number. Nonetheless, “… with 27 unique variants among 28 ST2-positive samples…”, “… with 19 unique variants among the 23 ST1-positive samples”, and “…with only six unique variants among 14 ST2-positive samples”: these “positive samples (28, 23, and 14 positive samples mentioned above)” could not be found in the table.

Moreover, “…relatively evenly distributed variants among samples”, “with only three unique variants”, and “… detected in more than five children” (lines 272 to 273): The statements “three unique variants” and “more than five children” were not found Table 5.

Lines 274 to 276: “the multiple unique variants of ST1 and ST2…” and “multiple variants of ST3 were only detected in a single sample”. The same issue as above, the sample information was absent.

Most importantly, it was easy to imagine that NGS generated massive amounts of data, and the authors tried to explain all the valuable information in this manuscript. However, the authors did not mention the reason for discussing unique intra-subtype variants within samples.

7.Lines 317 to 323: the paragraph should be moved to line 365, where the Blastocystis was discussed.

8.The authors tried to elucidate sub-assemblages and subtypes of G. duodenalis and Blastocystis sp. as possible, respectively. Intriguingly, detection of the sub-assemblages of G. duodenalis at least 6 rounds of PCR were performed, whereas NGS was conducted to determine the subtypes of Blastocystis sp.. It was noticed that the NGS was solely designed for Blastocystis sp. identification; why did the authors not conduct NGS to detect both protists, especially since they caused the most frequent co-infection in tested samples?

9.The NGS-based subtype determination of Blastocystis sp. was performed in this study. It is known that the protist might not be pathogenic or even beneficial for human hosts. Was it worth the effort to determine the subtypes?

PLOS authors have the option to publish the peer review history of their article (what does this mean?). If published, this will include your full peer review and any attached files.

Reviewer #1: No

Reviewer #2: No

---

## [Decision Letter · Decision Letter 1]

21 Nov 2024

Dear Dr Carmena,

We are pleased to inform you that your manuscript 'Asymptomatic carriage of intestinal protists is common in children in Lusaka Province, Zambia' has been provisionally accepted for publication in PLOS Neglected Tropical Diseases.

Best regards,

Luisa Magalhães

Academic Editor

Hira Nakhasi

Section Editor

Shaden Kamhawi

co-Editor-in-Chief

Paul Brindley

co-Editor-in-Chief

Reviewer's Responses to Questions

**Key Review Criteria Required for Acceptance?**

**Methods**

-Are the objectives of the study clearly articulated with a clear testable hypothesis stated?

-Is the study design appropriate to address the stated objectives?

-Is the population clearly described and appropriate for the hypothesis being tested?

-Is the sample size sufficient to ensure adequate power to address the hypothesis being tested?

-Were correct statistical analysis used to support conclusions?

-Are there concerns about ethical or regulatory requirements being met?

Reviewer #1: No additional comments.

Reviewer #2: (No Response)

**Results**

-Does the analysis presented match the analysis plan?

-Are the results clearly and completely presented?

-Are the figures (Tables, Images) of sufficient quality for clarity?

Reviewer #1: No additional comments.

Reviewer #2: (No Response)

**Conclusions**

-Are the conclusions supported by the data presented?

-Are the limitations of analysis clearly described?

-Do the authors discuss how these data can be helpful to advance our understanding of the topic under study?

-Is public health relevance addressed?

Reviewer #1: No additional comments.

Reviewer #2: (No Response)

**Editorial and Data Presentation Modifications?**

Reviewer #1: No additional comments.

Reviewer #2: (No Response)

**Summary and General Comments**

Reviewer #1: No comments.

Reviewer #2: (No Response)

PLOS authors have the option to publish the peer review history of their article (what does this mean?). If published, this will include your full peer review and any attached files.

Reviewer #1: No

Reviewer #2: No

---

## [Editor Report · Acceptance letter]

5 Dec 2024

Dear Dr Carmena,

We are delighted to inform you that your manuscript, "Asymptomatic carriage of intestinal protists is common in children in Lusaka Province, Zambia," has been formally accepted for publication in PLOS Neglected Tropical Diseases.

Best regards,

Shaden Kamhawi

co-Editor-in-Chief

Paul Brindley

co-Editor-in-Chief
